



# Inter-comparison of wind measurements in the atmospheric boundary layer with Aeolus and a ground-based coherent Doppler lidar network over China

Songhua Wu[1,2,3], Kangwen Sun[1], Guangyao Dai[1], Xiaoye Wang[1], Xiaoying Liu[1], Bingyi Liu[1,2], Xiaoquan Song[1,3], Oliver Reitebuch[4], Rongzhong Li[5], Jiaping Yin[5], Xitao Wang[5]

[1] College of Marine Technology, Faculty of Information Science and Engineering, Ocean University of China, Qingdao, 266100, China
[2] Laboratory for Regional Oceanography and Numerical Modelling, Pilot National Laboratory for Marine Science and Technology (Qingdao), Qingdao, 266200, China
[3] Institute for Advanced Ocean Study, Ocean University of China, Qingdao, 266100, China
[4] German Aerospace Center (Deutsches Zentrum für Luft- und Raumfahrt e.V., DLR), Institute of Atmospheric Physics, Oberpfaffenhofen, 82234, Germany
[5] Qingdao Leice Transient Technology Co., Ltd., Qingdao, 266100, China

*Correspondence to*: Songhua Wu (wush@ouc.edu.cn)

**Abstract.** After the successful launch of Aeolus which is the first spaceborne wind lidar developed by the European Space Agency (ESA) on 22 August 2018, we deployed several ground-based coherent Doppler wind lidars (CDLs) to verify the wind observations from Aeolus. By the simultaneous wind measurements with CDLs at 17 stations over China, the Rayleigh-clear and Mie-cloudy horizontal-line-of-sight (HLOS) wind velocities from Aeolus in the atmospheric boundary layer are compared with that from CDLs. To ensure the quality of the measurement data from CDL and Aeolus, strict quality controls are applied in this study. Overall, 52 simultaneous Mie-cloudy comparison pairs and 387 Rayleigh-clear comparison pairs from this campaign are acquired. All of the Aeolus-produced L2B Mie-cloudy HLOS, Rayleigh-clear HLOS and CDL-produced HLOS are compared individually. For the inter-comparison result of Mie-cloudy HLOS wind and CDL-produced HLOS wind, the correlation coefficient, the standard deviation, the scaled MAD and the bias are 0.83, 3.15 m·s⁻¹, 2.64 m·s⁻¹ and -0.25 m·s⁻¹ respectively, while the "y=ax" slope, the "y=ax+b" slope and the "y=ax+b" intercept are 0.93, 0.92 and -0.33 m·s⁻¹. For the Rayleigh-clear HLOS wind, the correlation coefficient, the standard deviation, the scaled MAD and the bias are 0.62, 7.07 m·s⁻¹, 5.77 m·s⁻¹ and -1.15 m·s⁻¹ respectively, while the "y=ax" slope, the "y=ax+b" slope and the "y=ax+b" intercept are 1.00, 0.96 and -1.2 m·s⁻¹. It is found that the standard deviation, the scaled MAD and the bias on ascending tracks are slightly better than that on descending tracks. Moreover, to evaluate the accuracy of Aeolus HLOS wind measurements under different product baselines, the Aeolus L2B Mie-cloudy HLOS wind data and L2B Rayleigh-clear HLOS wind data under Baselines 07/08, Baselines 09/10, and Baseline 11 are compared against the CDL-retrieved HLOS wind data separately. From the comparison results, marked misfits between the wind data from Aeolus Baselines 07/08 and wind data from CDL in planetary boundary layer are found. With the continuous calibration and validation and product processor updates, the performances of



Aeolus wind measurements under Baselines 09/10 and Baseline 11 are improved significantly. Considering the influence of turbulence and convection in the planetary boundary layers, higher values for the vertical velocity are common in this region.
Hence, as a special note, the vertical velocity could impact the HLOS wind velocity retrieval from Aeolus.

## 1 Introduction

Reliable instantaneous vertical profiling of the global wind field, especially over the Tropics and oceans, is crucial to many aspects of climate change, oceanography research, large-scale weather systems and weather prediction. It is also needed to address some of the key concerns of atmospheric dynamics and climate processes (Stoffelen et al., 2005). The wind field
measurements are important for studies of the large-scale monsoon systems and El Niño phenomenon in Tropics and jet stream in extra-tropics. Wind profiles are available from the global radiosonde network and from aircraft ascents and descents and from cruising altitudes for numerical weather prediction. However, due to the limitations of generally lacking over all ocean areas in the radiosonde network and wind observations only at a specific flight altitude (around 10-12 km about ground level) in aircraft measurements, a first ever spaceborne direct detection wind lidar, Aeolus, which is capable of providing the globally
high spatial and temporal vertical wind profiles is developed by European Space Agency (ESA) under the framework of Atmospheric Dynamics Mission (Stoffelen et al., 2005; ESA 1999; Reitebuch et al., 2012;). On 22 August 2018, the Aeolus was successfully launched onto its sun-synchronous orbit at a height of 320 km (Kanitz et al. 2018, Straume et al. 2018, Reitebuch et al. 2020). A quasi-global coverage is achieved daily (~16 orbits per day) and the orbit repeat cycle is 7 days (111 orbits). The orbit is sun-synchronous with a local equatorial crossing-time of 6 am/pm. The Atmospheric Laser Doppler
Instrument (ALADIN) is a direct detection high spectral resolution wind lidar operating at a laser wavelength of 354.8 nm and provides the vertical profiles of the Line-of-Sight (LOS) wind speed. In order to retrieve the LOS wind speeds, the Doppler shifts of light caused by the emotion of molecules and aerosol particles need to be identified. Aiming at this, a Fizeau interferometer is applied in the Mie channel to extract the frequency shift of the narrow-band particulate return signal by means of fringe imaging technique (Mckay, 2002). In the Rayleigh channel, two coupled Fabry-Perot interferometers are used to
analyze the frequency shift of the broad-band molecular return signal by the double edge technique (Chanin et al., 1989; Flesia and Korb, 1999).

After the successful launch of ALADIN, the data products have been released to the Aeolus Cal/Val teams on 16 December 2018. To recheck the quality of the data products, a validation of Aeolus winds by means of ground-based, airborne, shipborne reference instruments measurements are inevitable. From the validation campaigns conducted by German Aerospace Center
(Deutsches Zentrum für Luft- und Raumfahrt e.V., DLR), the wind observations from Aeolus and the well-validated ALADIN Airborne Demonstrator (A2D) are compared (Lux et al., 2020a; Witschas et al., 2020). An example of early validation of the Aeolus with a direct-detection Rayleigh-Mie Doppler lidar at Observatoire de Haute-Provence (OHP) in southern France (Khaykin et al., 2020). In November and December 2018, a unique validation of the wind products of Aeolus in the Atlantic Ocean west of the African continent was conducted by using the RV Polarstern cruise PS116 carried radiosondes (Baars et al.,





2020). In China, the wind observations from Aeolus are compared with the results from ground-based Radar wind profiler network and radiosonde over China (Guo et al., 2020; Liu et al., 2021). There are some significant validation campaigns as well in the worldwide (e.g., Bedka et al., 2020; Martin et al., 2020).

    As a member of the CAL/VAL teams, Ocean University of China (OUC) has performed one long-term observation campaigns with 1550 nm coherent Doppler wind lidar (CDL) all over China. During these campaigns, 439 simultaneous

measurement cases are acquired with the CDLs of types Wind3D 6000 and WindMast PBL, which are manufactured by Qingdao Leice Transient Technology Co., Ltd (http://www.leice-lidar.com/en/index.html). During the data processing, it was found that the atmospheric vertical velocity could influence the HLOS wind velocity measured by Aeolus in the planetary boundary layer. Hence, it should be specially noted that the HLOS wind velocities from CDL and Aeolus are different and should be corrected.

This paper provides the inter-comparison of the horizontal-line-of-sight (HLOS) wind velocities measured by CDL and Aeolus. The paper is organized as follows: in section 2 the simultaneous validation campaigns and the instrument deployed for the measurements are described. Section 3 presents the details to the inter-comparison strategy, the quality control and vertical velocity correction procedure. In section 4 we provide the HLOS wind velocity measurement examples and comparison results. Section 5 summaries the recent comparison results and compares those with ours.

**2 Validation campaigns in China and lidar introduction**

**2.1 Overview of the validation campaigns**

    Shortly after the successful launch of the Aeolus, the primary laser head FM-A (Flight Model – A) was switched on and an initial laser pulse energy of 65 mJ was achieved (Lux et al. 2020b). During 14 January and 14 February 2019, Aeolus was in standby-mode and switched-on with FM-A. After a final test with laser FM-A of Aeolus on 17 June, the transition to laser

FM-B took place. About half a year later, the validation campaign (VAL-OUC) performed by the Ocean University of China has been carried out since January 2020 at 17 stations. The comparison results of HLOS wind velocities in the atmospheric boundary layers from CDLs and Aeolus) are presented in section 4. The duration of the validation campaign (VAL-OUC) is from January to December 2020. The locations of the CDLs, the ascending and descending orbits of Aeolus are shown in Fig. 1. An overview and detailed information of the validation campaign are provided in Table 1.



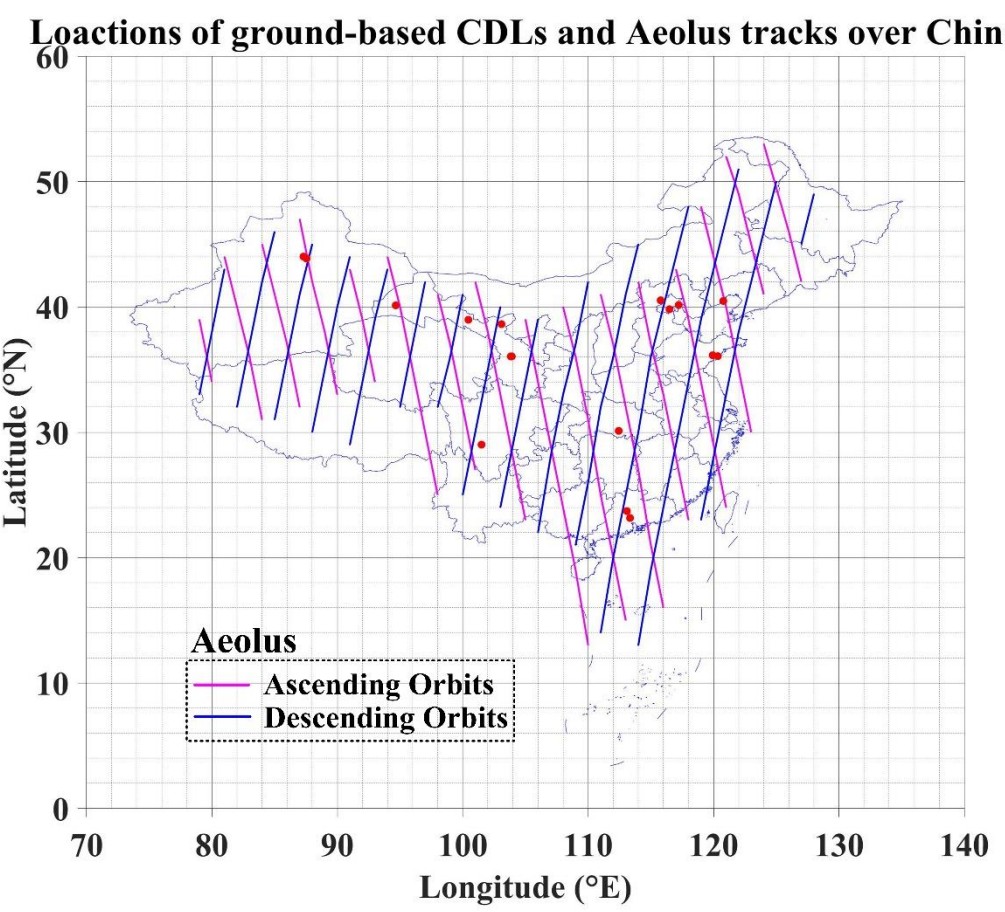

**Figure 1. Ground-based CDL observation sites of VAL-OUC campaign since January 2020. The pink and blue swath indicate the ascending and descending orbits of Aeolus. The red dots represent the locations of the CDLs.**

**Table 1. Overview of Aeolus validation campaigns performed by OUC**

| Validation campaigns | Instrument type | Measurement mode | Location | Latitude, Longitude, Altitude | Measurement period |
|---|---|---|---|---|---|
| VAL-OUC | WindMast PBL | DBS | Dunhuang | 40.12°N, 94.66°E 1.15 km | From 07 Jan. 2020 to 29 Dec. 2020 |
| | WindMast PBL | DBS | Lanzhou | 36.05°N, 103.91°E 1.51 km | From 07 Jan. 2020 to 29 Dec. 2020 |
| | WindMast PBL | DBS | Zhangye | 38.97°N, 100.45°E 1.46 km | From 05 Jan. 2020 to 27 Dec. 2020 |
| | Wind3D 6000 | DBS | Jingzhou | 30.11°N, 112.44°E 0.03 km | From 24 June 2020 to 22 July 2020 |
| | Wind3D 6000 | DBS | Pinggu, Beijing | 40.15°N, 117.22°E 0.05 km | From 21 Apr. 2020 to 02 June 2020 |
| | Wind3D 6000 | DBS | Changji | 44.01°N, 87.30°E | 03 Dec. 2020 |



| | | | 0.58 km | |
|---|---|---|---|---|
| Wind3D 6000 | DBS | Jiulong, Sichuan | 29.01°N, 101.50°E 2.90 km | From 24 Oct. 2020 to 29 Nov. 2020 |
| Wind3D 6000 | DBS | Jiaozhou, Shandong | 36.14°N, 119.93°E 0.02 km | 21 Dec. 2020 |
| Wind3D 6000 | DBS | Qingyuan, Guangdong | 23.71°N, 113.09°E 0.03 km | From 12 May 2020 to 27 Aug. 2020 |
| Wind3D 6000 | DBS | Xidazhuangke, Beijing | 40.52°N, 115.78°E 0.91 km | From 7 Jan. 2020 to 31 Mar. 2020 |
| Wind3D 6000 | DBS | Yizhuang, Beijing | 39.81°N, 116.48°E 0.04 km | From 07 Apr. 2020 to 25 Aug. 2020 |
| Wind3D 6000 | DBS | Huludao | 40.47°N, 120.78°E 0.10 km | From 01 Nov. 2020 to 28 Dec. 2020 |
| Wind3D 6000 | DBS | Wuwei | 38.62°N, 103.09°E 1.37 km | From 11 Apr. 2020 to 26 Dec. 2020 |
| Wind3D 6000 | DBS | Lanzhou | 36.05°N, 103.83°E 1.53 km | From 04 Jan. 2020 to 26 Dec. 2020 |
| Wind3D 6000 | DBS | South China University of Technology | 23.16°N, 113.34°E 0.03 km | From 13 Oct. 2020 to 29 Dec. 2020 |
| Wind3D 6000 | DBS | Urumqi | 43.85°N, 87.55°E 0.84 km | From 14 Oct. 2020 to 24 Dec. 2020 |
| Wind3D 6000 | DBS | Qingdao | 36.07°N, 120.34°E 0.04 km | From 02 Nov. 2020 to 28 Dec. 2020 |

## 2.2 The ALADIN and CDL descriptions

In this subsection, the unique payload of Aeolus, the Atmospheric Laser Doppler Instrument (ALADIN), and the ground-based reference coherent Doppler wind lidar are briefly described.

### 2.2.1 ALADIN

ALADIN is a direct detection high spectral resolution wind lidar which operates at the wavelength of 354.8 nm with a laser pulse energy around 65mJ and with a repetition of 50.5 Hz (Lux et al. 2020b). It is equipped with a 1.5 m diameter telescope

to collect the backscatter light from molecules and aerosol particles. The high spectral resolution design of ALADIN allows for the simultaneous detection of the molecular (Rayleigh) and particle (Mie) backscattered signals in two separate channels, each sampling the wind in 24 vertical height bins with a vertical range resolution between 0.25 km and 2.0 km. This makes it possible to deliver winds both in clear and (partly) cloudy conditions down to optically thick clouds at the same time. The horizontal resolution of the wind observations is about 90 km for the Rayleigh channel and about 10-15 km for the Mie channel.

A detailed description of the instrument design and a demonstration of the measurement concept are introduced in e.g. (Reitebuch et al., 2009; 2012; Straume et al., 2018; ESA 2008; Marksteiner 2013).



The data products of Aeolus are processing at different levels mainly including Level 0 (instrument housekeeping data), Level 1B (engineering-corrected HLOS winds), Level 2A (aerosol and cloud layer optical properties), Level 2B (meteorologically-representative HLOS winds) and Level 2C (Aeolus-assisted wind vectors from ECMWF model) (Tan et al., 110   2008; Rennie et al., 2020a). In this study, the Level 2B HLOS wind velocities are used. Within the Level 2B processor, the Rayleigh-clear and Mie-cloudy winds are classified and the temperature and pressure correction are applied for the Rayleigh wind retrieval.

### 2.2.2 Coherent Doppler wind lidar instrument

Lidar is one of the most accurate optical remote sensing techniques for wind field measurements. The 1550 nm wavelength 115   all-fiber Coherent Doppler wind Lidar (CDL) with high resolution takes advantage of the fact that the frequency of the echo signal is shifted from the local-oscillator light because of the Doppler effect which occurs from backscattering of aerosols. The Doppler shift in the frequency of the backscattered signal is analyzed to obtain the LOS velocity along the lidar beam direction. The CDL is based on the heterodyne technique, consisting of a single frequency seed laser source, an acousto-optic modulator, an Erbium doped fiber amplifier, optical isolators and amplified spontaneous emission noise filters, an optical switch, a 120   transceiver telescope, a balanced detector and an analog-to-digital converter and a Fast Fourier Transform signal processor. Further information regarding the CDL is described in a separate paper (Wu et al., 2016).

The CDL of types Wind3D 6000 and WindMast PBL are lidar systems for wind measurements in the lower atmosphere. The devices are developed by the Leice Transient Technology and designed with consideration for the needs of the meteorological application, wind energy industry and aviation safety. The specifications of the CDLs are listed in Table 2.

**Table 2. Overview of CDL specifications used for Aeolus validation**

| Qualification | Specifications | |
| --- | --- | --- |
| | Wind3D 6000 | WindMast PBL |
| Wavelength | 1550 nm | 1550 nm |
| Repetition rate | 10 kHz | 10 kHz |
| Pulse energy | 160 uJ | 100 uJ |
| Pulse width | 100 ns to 400 ns | 100 ns to 400 ns |
| Detection range | 80 m to 6000 m | 30 m to 4000 m |
| Data update rate | 4 Hz | 4 Hz |
| Range resolution | 15 m to 60 m | 15 m to 30 m |
| Wind speed accuracy | $\leqslant 0.1$ $m \cdot s^{-1}$ | $\leqslant 0.1$ $m \cdot s^{-1}$ |
| Wind speed range | $\pm 75$ $m \cdot s^{-1}$ | $\pm 75$ $m \cdot s^{-1}$ |
| Wind direction accuracy | $0.1°$ | $0.1°$ |

To evaluate the accuracy and precision of ground-based CDL measurements, the Wind3D 6000 and the WindMast PBL were validated with mast mounted cup anemometers and wind vanes at Haiyang, Shandong Province of China from 23 July 2021 to 30 July 2021. The measurement heights selected for comparison are 50 m, 100m. The comparison results for Wind3D





6000 and WindMast PBL are shown in Fig. 2 (a) and (b), respectively. By performing ordinary least squares linear regressions
of the CDLs and cup anemometers wind measurements, the slopes, offsets, standard deviations and correlation coefficients are
acquired and they are within the acceptable limits. The statistic results of the validation are shown in Table 3. Hence, the CDL
of types Wind3D 6000 and WindMast PBL can be act as reference instruments for the validation of Aeolus in the atmospheric
boundary layer.

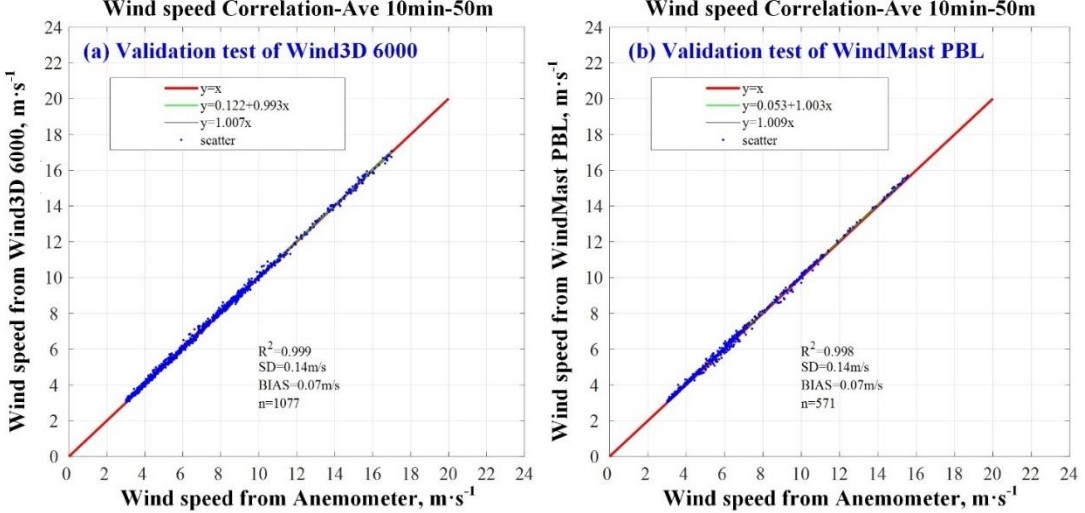

**Figure 2. Evaluation tests of (a) Wind3D 6000 and (b) WindMast PBL performance by comparing their measurements against the conventional wind measurements with mast mounted cup anemometers and wind vanes.**

**Table 3. Statistical results of the validation tests between Wind3D 6000/WindMast PBL and anemometers.**

| Specifications | Wind3D 6000 | WindMast PBL |
|---|---|---|
| N points | 1077 | 571 |
| Correlation | 0.999 | 0.998 |
| SD ( $m \cdot s^{-1}$ ) | 0.14 | 0.14 |
| BIAS ( $m \cdot s^{-1}$ ) | 0.07 | 0.07 |
| "y=ax" Slope | 1.007 | 1.009 |
| "y=ax+b" Slope | 0.993 | 1.003 |
| "y=ax+b" Intercept ( $m \cdot s^{-1}$ ) | 0.122 | 0.053 |

## 3 Inter-comparison of Aeolus and CDL measurements

During the validation campaigns of VAL-OUC, the wind field measurements at the sites over China are continuously
performed, except during the period of the CDL maintenances.





## 3.1 Inter-comparison strategy

In Fig. 3, we provide the flowchart of the comparison between Wind3D 6000/WindMast PBL measurements against Aeolus measurements. To ensure the quality of the measurement data from Wind3D 6000 and WindMast PBL, we only used the CDL data with SNR>-10 dB. For Aeolus, only observations with the corresponding "validity flag" of TRUE are considered. For the

comparison, only the Mie-cloudy and Rayleigh-clear wind velocities from the L2B product with estimated errors lower than 4 m/s and 8 m/s, respectively, are selected. Moreover, the Aeolus lowest atmospheric bins close to the ground are also removed from the comparison because lowest atmospheric range bins from Aeolus could be contaminated with ground. In this study, the horizontal separations between the locations of CDLs and Aeolus measurement ground track should be less than 80 km. Since the CDL provide continuous atmospheric observations, there is no time difference between CDL and simultaneous

Aeolus measurements. Vertical averaging of the CDL measurements over 1 Aeolus range bin is also performed.

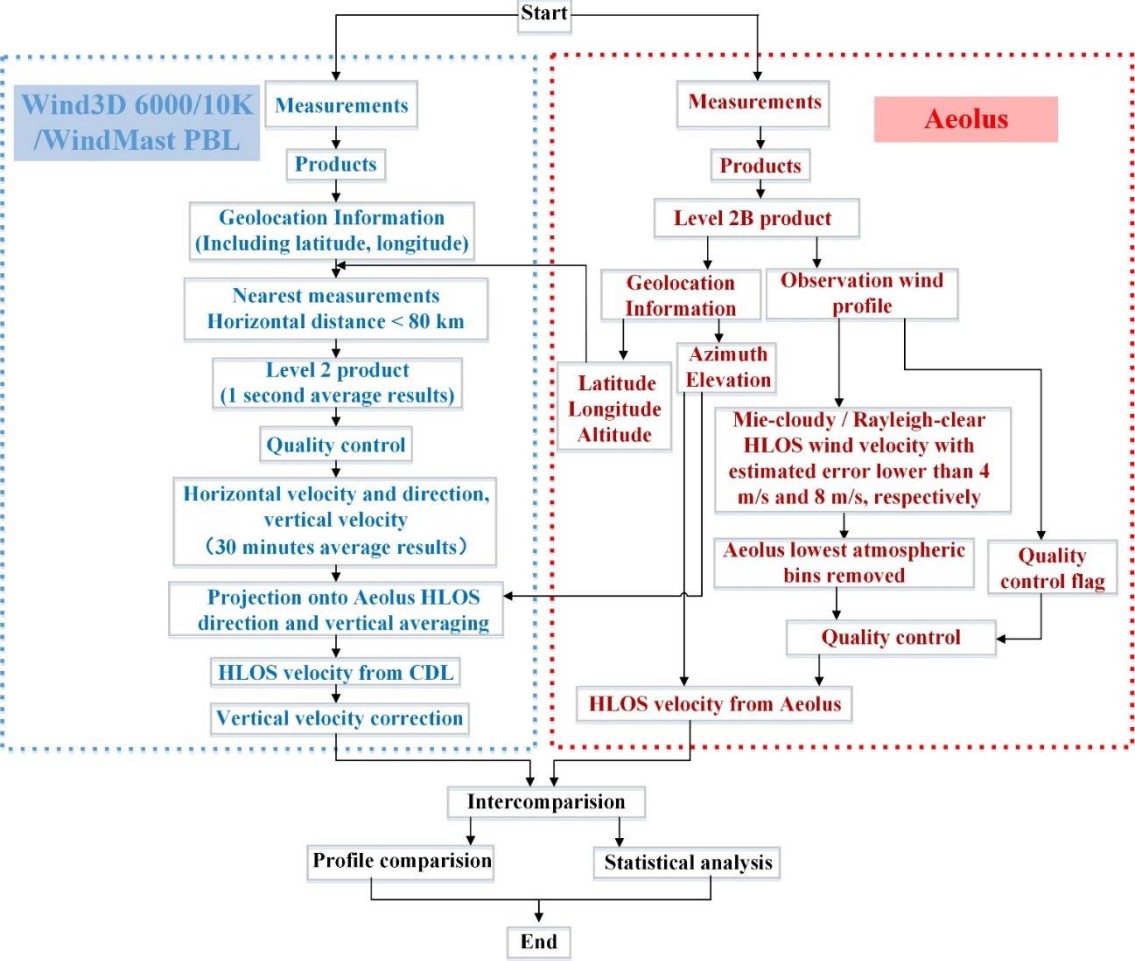

**Figure 3. Sketch of the comparison between CDL and Aeolus in the atmospheric boundary layer.**





During the measurement procedure, to observe the three-dimensional wind speed and direction, the Doppler Beam Swing (DBS) scanning mode of CDLs is applied. The five-beam DBS scanning technique is mainly used to retrieve the wind profiles

by measuring the LOS wind speeds in the vertical, the north, the east, the south, and the west directions. The original wind product of CDL is one second average results (Level 2 product). By considering the low horizontal spatial resolution of Aeolus data (about 90 km for the Rayleigh-clear wind velocities and 10 km for the Mie-cloudy wind velocities), 30 minutes ($\pm$15 minutes) average of CDL wind product is applied and the nearest observations profile provided by CDL and Aeolus is selected by using the geolocation information in each measurement case. Besides, since Aeolus can only deliver the HLOS winds data,

the simultaneous wind measurements from CDL have to be projected onto the Aeolus HLOS direction using the azimuth angle from Aeolus. The CDL-HLOS wind ($HLOS_{CDL}$) is calculated as

$$HLOS_{CDL} = V_{CDL-EW} \cdot \sin(Azi_{Aeolus}) + V_{CDL-SN} \cdot \cos(Azi_{Aeolus}) . \tag{1}$$

$V_{CDL-EW}$, $V_{CDL-SN}$ are the east-west wind speed and the south-north wind speed measured by CDL respectively, $Azi_{Aeolus}$ is the azimuth angle of ALADIN provided by Aeolus products.

**3.2 Influence of vertical velocity in the boundary layer**

In planetary boundary layer, the vertical velocity of air mass has pronounced impact on the HLOS wind velocity measured by Aeolus. The schematic diagram of the vertical velocity impacts on the HLOS velocity retrieval is presented in Fig. 4. Hence, the difference between the HLOS wind velocities from CDL and Aeolus should be specially noticed during the data processing.

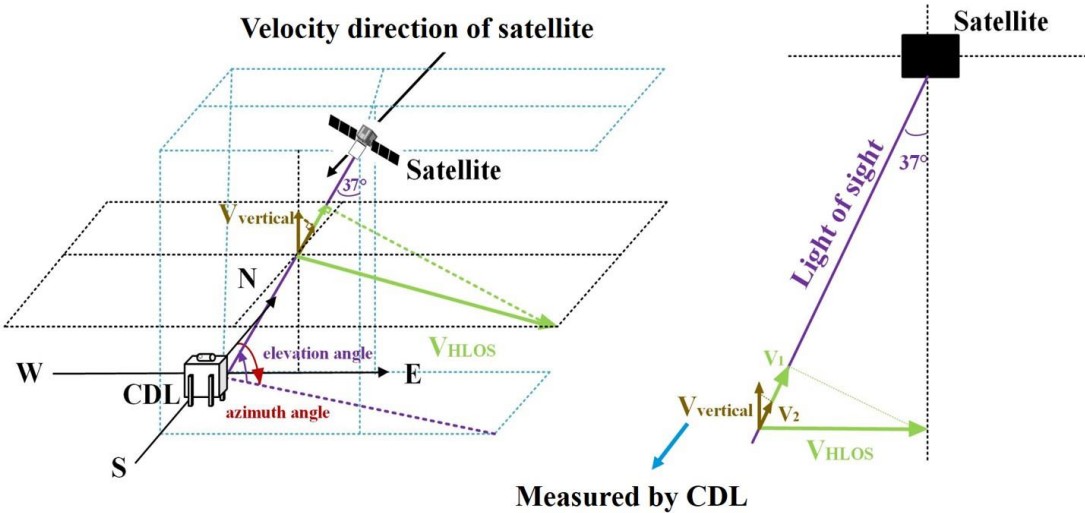


**Figure 4. The schematic diagram of the vertical velocity impact on the HLOS velocity retrieval of Aeolus.**

If the projection of $V_{vertical}$ onto $HLOS_{Aeolus}$ is in the same direction as $HLOS_{Aeolus}$, the relationship between $HLOS_{CDL}$ and $HLOS_{Aeolus}$ should be expressed as





$$HLOS_{CDL} = HLOS_{Aeolus} - V_{vertical} \cot 37°. \qquad (2)$$

Otherwise, if the projection of $V_{vertical}$ onto $HLOS_{Aeolus}$ is in the opposite direction as $HLOS_{Aeolus}$, the relationship between $HLOS_{CDL}$ and $HLOS_{Aeolus}$ should be changed to

$$HLOS_{CDL} = HLOS_{Aeolus} + V_{vertical} \cot 37°. \qquad (3)$$

Vertical wind measurements in the planetary boundary layer during the validation campaigns at each site are performed. In Fig. 5, one vertical wind measurement case with a temporal resolution of 30 minutes is provided. From this figure, it is found

that the typical temporal average of vertical wind is $\pm 0$ m·s$^{-1}$ to $\pm 0.40$ m·s$^{-1}$. According to the schematic diagram of the vertical velocity impact plotted in Fig. 4, the vertical wind with a speed of $\pm 0.40$ m·s$^{-1}$ will introduce an error of $\pm 0.53$ m·s$^{-1}$ in retrieving HLOS. Thus, the HLOS from CDL is corrected considering the vertical velocity effect.

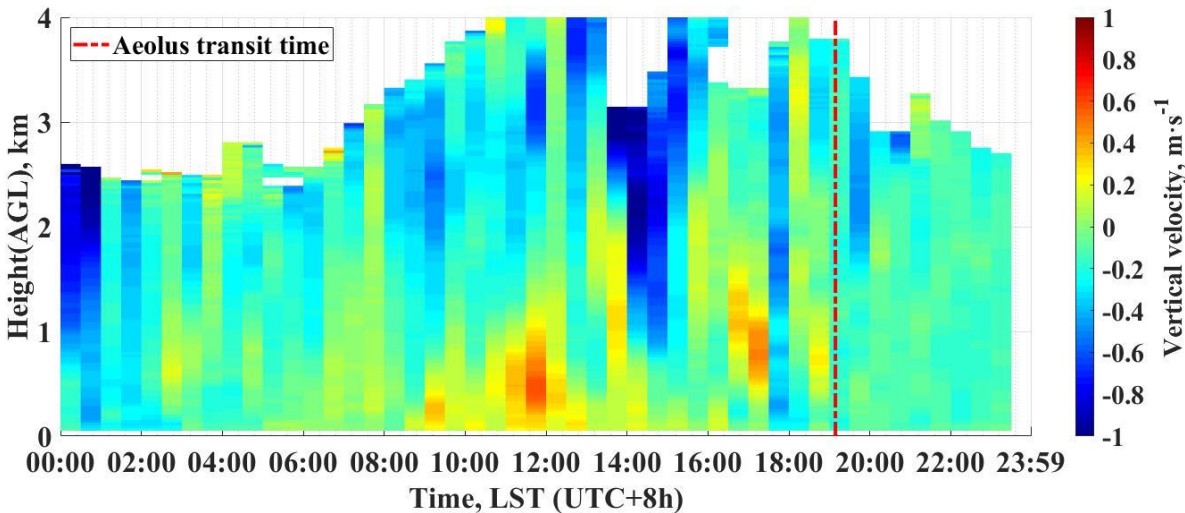

**Figure 5. Vertical velocity measured on 10 May 2020 in Zhangye, China. The red dashed line indicates the Aeolus transit time.**

## 4. Results and discussion

In the validation campaign, the CDLs of types Wind3D 6000 and WindMast PBL are deployed at different observations sites. According to the sketch illustrated in Fig. 3, the measurement data from CDLs and Aeolus are processed. In this section, some

examples and a statistical analysis is presented.



## 4.1 Profiles comparison

In Fig. 6, a measurement case of wind field observed with Wind3D 6000 in Zhangye, Gansu Province on 10 May 2020 is provided. In this figure, the Aeolus Mie-cloudy HLOS wind velocity (Fig. 6 (a)) and Rayleigh-clear HLOS wind velocity (Fig. 6 (b)) on 10 May 2020 are shown. The red dashed lines in Fig. 6 (a) and (b) show the nearest observation profiles on this orbit to Zhangye. Meanwhile, the profiling of the SNR (Fig. 6 (c)), wind velocity (Fig. 6 (d)), wind direction (Fig. 6 (e)) and vertical velocity (Fig. 6 (f)) measured by Wind3D 6000 in the planetary boundary layer are presented with a temporal resolution of 1 minute. The red dashed lines indicate the Aeolus transit time and the Aeolus overflight was around 19:09 LST (11:09 UTC).







Figure 6. Aeolus-retrieved (a) Mie HLOS velocity and (b) Rayleigh HLOS velocity and profiling of (c) SNR, (d) wind velocity, (e) wind direction and (f) vertical velocity in the planetary boundary layer measured by CDL on 10 May 2020 in Zhangye, China. The red dashed lines in (a) and (b) denote the location of Zhangye, and the red dashed lines in (c), (d), (e) and (f) indicate the Aeolus transit time over Zhangye.



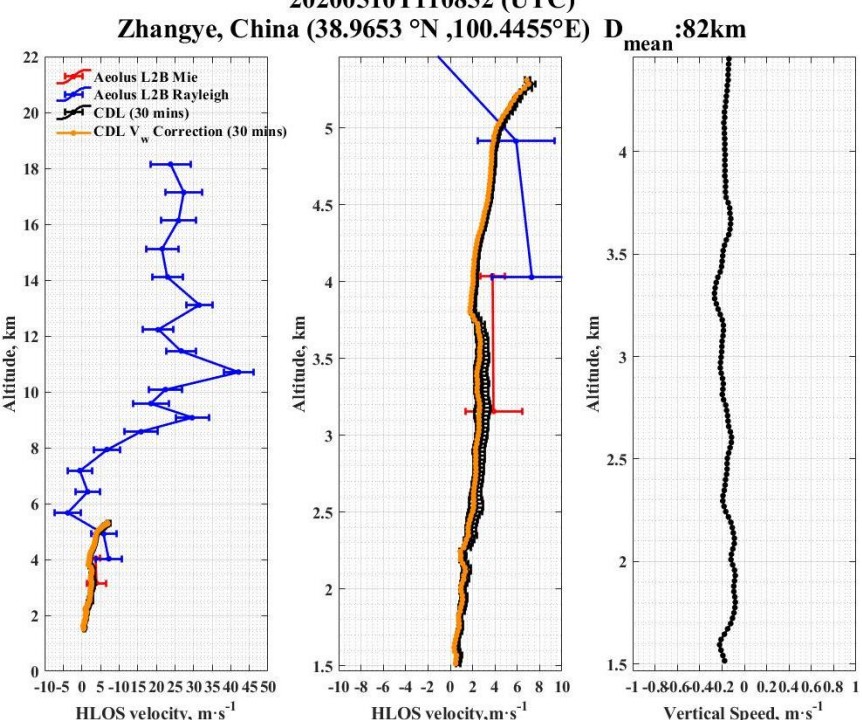


**Figure 7. Inter-comparison of HLOS wind velocities measured with CDL and Aeolus on 10 May 2020 at Zhangye, China. The left panel is the overall view of the inter-comparison result, where the red line represents the Aeolus L2B Mie-cloudy HLOS wind profile; the blue line represents the Aeolus L2B Rayleigh-clear HLOS wind profile; the black line represents the CDL-retrieved HLOS wind profile while the yellow line represents the vertical velocity corrected**
**CDL-retrieved HLOS wind profile. The middle panel shows the partly view of the inter-comparison result, the lines are the same as the left panel. The right panel is the vertical velocity profile.**

To compare the measurement results in Zhangye, the simultaneous profiles of HLOS wind velocities observed with Aeolus and Wind3D 6000 are provided in Fig. 7. In this figure, the CDL-retrieved HLOS wind velocities with and without vertical velocity correction are compared against the Aeolus Mie L2B products and Rayleigh L2B products. The vertical velocity

profile is plotted as well. From this figure, it is found that the Aeolus L2B Rayleigh-clear HLOS products in the planetary boundary layer are not always available but are trustable when they are provided. The Aeolus L2B Mie-cloudy HLOS in the planetary boundary layer fit well with the synchronous CDL measurements. Additionally, the 30-minute averaged vertical velocity profile shows that the vertical velocity is around -0.16 $m \cdot s^{-1}$, which could introduce the error of -0.21 $m \cdot s^{-1}$ according to the method provided in Section 3.2.




**Figure 8. Inter-comparison of HLOS wind velocities measured with CDL and Aeolus at (a) Xidazhuangke (Beijing), (b) Wuwei (Gansu Province), (c) Huludao (Liaoning Province) and (d) Qingdao (Shandong Province) on 21 January, 18 September, 15 November and 16 November 2020, respectively. The lines are the same as those of Fig. 7.**

To compare the measurement results, four simultaneous profiles of HLOS wind velocities observed with Aeolus and CDL on 21 January, 18 September, 15 November and 16 November 2020 (UTC) at Xidazhuangke (Beijing), Wuwei (Gansu Province), Huludao (Liaoning Province) and Qingdao (Shandong Province) are shown in Fig. 8. In this figure, the CDL-retrieved HLOS wind velocities with and without vertical velocity correction are compared against the Aeolus L2B Mie-cloudy products and Rayleigh-clear HLOS products. The vertical velocity profiles are shown as well. From Fig. 8 (a), it is found that

the Aeolus L2B Mie-cloudy products in the planetary boundary layer fit well with the CDL-retrieved HLOS wind velocities. In Fig. 8 (b), (c) and (d), the CDL-retrieved HLOS wind velocities and the Aeolus L2B Rayleigh-clear HLOS products agree





well from the planetary boundary layer to the altitude of around 4 km while the CDL-retrieved profile are all in the range of Aeolus estimated errors. Besides, in the inter-comparison case of Huludao (Fig. 8 (c)), in the altitude of around 1.2 km to 1.7 km, the vertical velocity measured by CDL is larger than 1.00 $m \cdot s^{-1}$, which could produce the error of about 1.33 $m \cdot s^{-1}$. The

vertical velocity corrected results (the yellow line) show the better agreement with the Aeolus L2B Rayleigh-clear HLOS wind velocities than the original CDL-retrieved HLOS wind velocities.

**4.2 Statistics comparison**

In this section, we compare the HLOS wind velocity results from Aeolus observations with the accompanying ground-based CDLs measurements. During the time period of January to December 2020 within the VAL-OUC campaign, 52 simultaneous

Mie-cloudy comparison pairs and 387 Rayleigh-clear comparison pairs at 17 stations are acquired. Figure 9 shows the counts of the comparison data pairs at different detection height ranges of Mie-cloudy channel and Rayleigh-clear channel. It can be seen that the heights of the comparison pairs are mainly in and close to the planetary boundary layer.

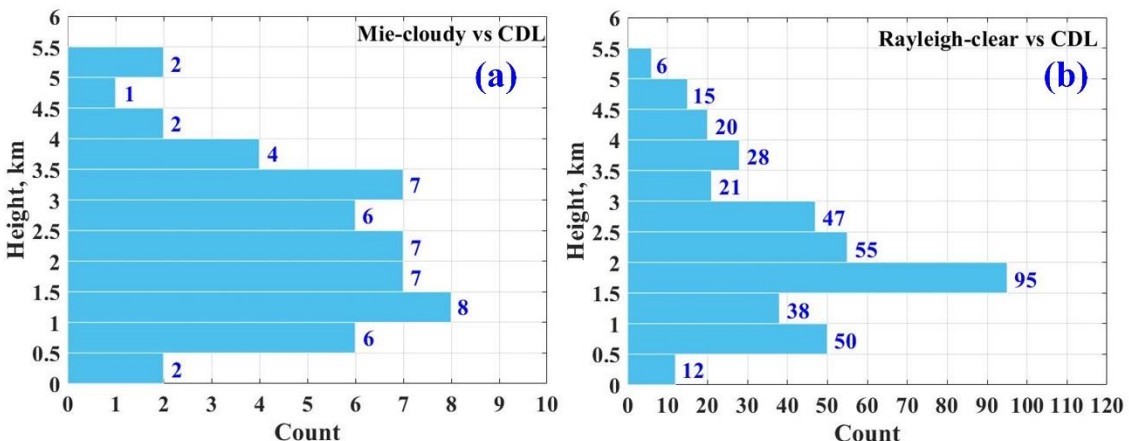

**Figure 9. Counts of data pairs at different height ranges of (a) Mie-cloudy vs CDL and (b) Rayleigh-clear vs CDL.**

In Fig. 10, the Mie-cloudy HLOS wind velocities and Rayleigh-clear HLOS wind velocities from Aeolus are compared with that from CDL, respectively. Figure 10(a) presents the scatter diagram of Aeolus L2B Mie-cloudy HLOS and CDL HLOS. 52 measurement cases for Mie−cloudy winds are available for the comparison. From this result, the correlation coefficient, the standard deviation, the scaled MAD, the bias are 0.83, 3.15 $m \cdot s^{-1}$, 2.64 $m \cdot s^{-1}$ and -0.25 $m \cdot s^{-1}$ respectively, while the "y=ax" slope, "y=ax+b" slope and "y=ax+b" intercept are 0.93, 0.92 and -0.33 $m \cdot s^{-1}$. In Fig. 10 (c), the scatter diagram of Aeolus

L2B Rayleigh-clear HLOS and CDL HLOS data are plotted. There are 387 comparisons are taken into consideration. Accordingly, the correlation coefficient, the standard deviation, the scaled MAD, and the bias are 0.62, 7.07 $m \cdot s^{-1}$, 5.77 $m \cdot s^{-1}$, -1.15 $m \cdot s^{-1}$ respectively, while the "y=ax" slope, "y=ax+b" slope and "y=ax+b" intercept are 1.00, 0.96 and -1.20 $m \cdot s^{-1}$. Table 4 summarizes the statistical results of the comparison. It should be emphasized that before these comparisons process, the data with HLOS differences larger than the original standard deviation are removed and are not considered. For Mie-cloudy HLOS





wind and Rayleigh-clear HLOS wind, 15 (22.39%) comparison pairs and 94 (19.54%) comparison pairs are removed respectively. Figure 10 (b) and (d) show statistic histograms of the count comparison between CDL-retrieved HLOS wind with Aeolus Mie-cloudy HLOS wind and between CDL-retrieved HLOS wind with Aeolus Rayleigh-clear HLOS wind.

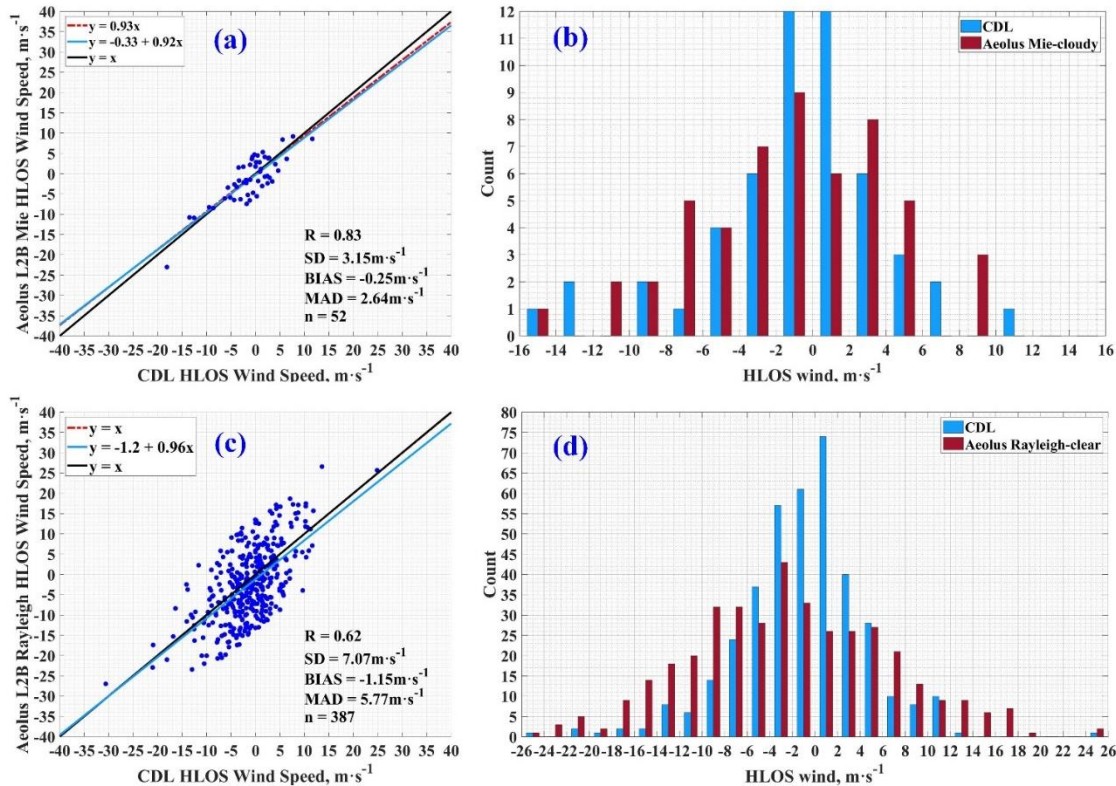

**Figure 10. Comparisons of Aeolus L2B Rayleigh-clear HLOS wind velocities and Mie-cloudy HLOS wind velocities against that from CDL. In Fig. 10 (a) and (c), the red dotted lines represent the "y=ax" fitting lines; the blue lines represent the "y=ax+b" fitting lines; the black lines represent the "y=x" reference line. Figure 10 (b) and (d) show the histogram of counts of HLOS wind velocities, where the blue columns represent the count of CDL HLOS wind velocities and the red columns represent the count of Aeolus HLOS wind velocities.**

**Table 4. Statistical comparison of Aeolus HLOS winds and CDL-retrieved HLOS winds.**

| Channel | Mie-cloudy | Rayleigh-clear |
|---|---|---|
| N points | 52 | 387 |
| Correlation | 0.83 | 0.62 |
| SD ( $m \cdot s^{-1}$ ) | 3.15 | 7.07 |
| Scaled MAD ( $m \cdot s^{-1}$ ) | 2.64 | 5.77 |
| BIAS ( $m \cdot s^{-1}$ ) | -0.25 | -1.15 |
| "y=ax" Slope | 0.93 | 1.00 |
| "y=ax+b" Slope | 0.92 | 0.96 |
| "y=ax+b" Intercept ( $m \cdot s^{-1}$ ) | -0.33 | -1.20 |






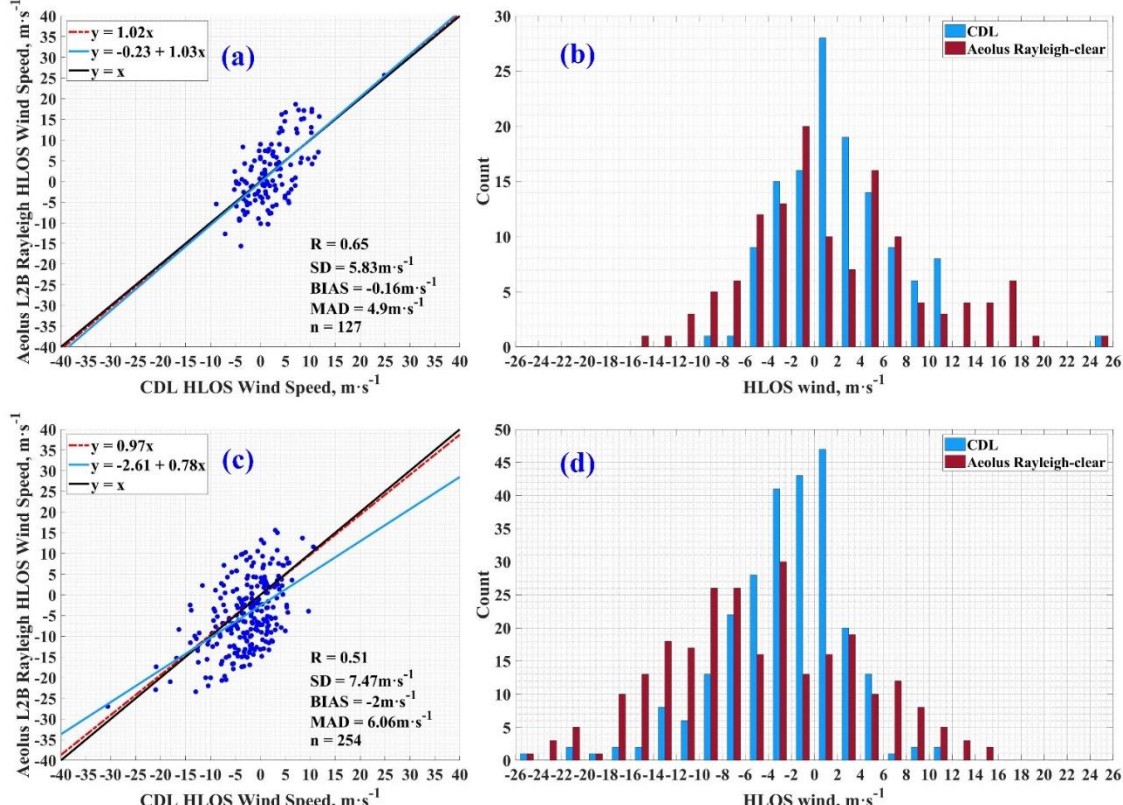

**Figure 11. Comparisons of Aeolus Rayleigh-clear HLOS against the CDL-retrieved HLOS according to the measurements made on (a)(b) ascending and (c)(d) descending tracks. The lines and the histograms represent the same as those of Fig. 10.**

**Table 5. Statistical comparison of Aeolus Rayleigh ascending/descending HLOS winds and CDL-retrieved HLOS winds.**

| Ascending/Descending | Ascending | Descending |
|---|---|---|
| N points | 127 | 254 |
| Correlation | 0.65 | 0.51 |
| SD ( $m \cdot s^{-1}$ ) | 5.83 | 7.47 |
| Scaled MAD ( $m \cdot s^{-1}$ ) | 4.90 | 6.06 |
| BIAS ( $m \cdot s^{-1}$ ) | -0.16 | -2.00 |
| "y=ax" Slope | 1.02 | 0.97 |
| "y=ax+b" Slope | 1.03 | 0.78 |
| "y=ax+b" Intercept ( $m \cdot s^{-1}$ ) | -0.23 | -2.61 |

Additionally, the scatter plots and the statistics histograms of Aeolus Rayleigh-clear HLOS against the CDL-retrieved HLOS according to the measurements made on ascending and descending tracks are presented individually in Fig. 11. Figure 11 (a) indicates the comparison between the Aeolus ascending measurements against that from CDL. It is found that the correlation



coefficient, the standard deviation, the scaled MAD and the bias are 0.65, 5.83 m·s$^{-1}$, 4.90 m·s$^{-1}$, -0.16 m·s$^{-1}$, respectively, while the "y=ax" slope, "y=ax+b" slope and "y=ax+b" intercept are 1.02, 1.03 and -0.23 m·s$^{-1}$. Figure 11 (b) shows the

comparison between the Aeolus descending measurements against that from CDL. The correlation coefficient, the standard deviation, the scaled MAD and the bias are 0.51, 7.47 m·s$^{-1}$, 6.06 m·s$^{-1}$ and -2.00 m·s$^{-1}$, respectively. Besides, the "y=ax" slope, "y=ax+b" slope and "y=ax+b" intercept are 0.97, 0.78 and -2.61 m·s$^{-1}$. Consequently, the standard deviation, the scaled MAD and the bias on ascending tracks are slightly better than that on descending tracks. The statistic results are summarized in Table 5.

Since the duration for the comparison of Aeolus and CDL synchronous measurements lasted during 2020, the baselines of the Aeolus product changed accordingly during this period (Rennie et al., 2020b). From baseline 07 to baseline 08 of the L2B wind product processor, the associated new auxiliary file carrying the parameters for needed for the M1 mirror temperature correction were provided, but not used in the L2B processing. After the deployment of baseline 09, the new auxiliary file with the M1 mirror temperature correction parameters were used, thus correcting for the associated biases in the L2B wind product.

The baseline 10 dataset consists of baseline 09 data from April 2020 to October 2020 and the FM-B low bias reprocessed dataset of 2019. With the Baseline 11 processor deploying, different SNR thresholds for classification of Mie and Rayleigh and an option to transfer Mie SNR results to the Rayleigh channel were added, which allows to do SNR based classification for the Rayleigh channel, resulting in a clear quality improvement. Therefore, to evaluate the impact of updating baseline products on the HLOS measurements, the Aeolus L2B Mie-cloudy HLOS data is compared with the corresponding CDL

HLOS data in Fig. 12 (a), meanwhile Rayleigh-clear HLOS data from Baseline 07 and 08, Baseline 09 and 10, and Baseline 11 are compared against the CDL-retrieved individually in Fig. 12 (b), (c) and (d).

During the comparison period, the Aeolus L2B HLOS measurements between January and April 2020 are produced with the baseline 07 and 08, the measurements between May and September 2020 are with the baseline 09 and 10, and the rest measurements from October 2020 are supported by Baseline 11.In Fig. 12 (a), the scatter plot for the Aeolus L2B Rayleigh-

clear HLOS with the Baselines of 07 and 08 is provided, from where it can be found that the correlation coefficient is 0.39 and the standard deviation, the scaled MAD and the bias are 10.20 m·s$^{-1}$, 8.42 m·s$^{-1}$ and -1.23 m·s$^{-1}$, respectively. Hence the Aeolus products with Baseline 07 and 08 should be calibrated furtherly. From the contrast results shown in Fig. 12 (c) and (e), thanks to the M1 mirror temperature correction from baseline 08 processor to baseline 09 processor and continuous calibration and validation activities carried out by the CAL/VAL team of Aeolus, the correlation coefficients, the standard deviations, the

scaled MAD and the biases are significantly improved than that from Baselines 07/08. The correlation coefficient reaches to 0.75 (0.86) for scatter plot with Baselines 09/10 (Baseline 11). The corresponding standard deviation and scaled MAD decreases to 4.66 m·s$^{-1}$ (4.76 m·s$^{-1}$) and 3.84 m·s$^{-1}$ (3.91 m·s$^{-1}$), and the bias is suppressed to -0.98 m·s$^{-1}$ (-0.13 m·s$^{-1}$) during the comparison with Baselines 09/10 (Baseline 11). The statistical comparison results are also presented in Table 6.





From Fig. 12(b), (d) and (f), the count histograms of comparison also show the significant improvement of the comparison
results from baseline 07/08 to baseline 09/10 and baseline 11.

**Figure 12. The comparison between the Aeolus L2B Rayleigh HLOS data from (a)(b) Baseline 07 and 08, (c)(d) Baseline 09 and 10, and (e)(f) Baseline 11 against the CDL-retrieved HLOS data. The lines and the histograms represent the same as those of Fig. 10.**





**Table 6. Statistical comparison of Aeolus Rayleigh HLOS winds of different Baselines and CDL-retrieved HLOS winds.**

| Baselines | 07 and 08 | 09 and 10 | 11 |
|---|---|---|---|
| N points | 156 | 106 | 100 |
| Correlation | 0.39 | 0.75 | 0.86 |
| SD ( m·s$^{-1}$ ) | 10.20 | 4.66 | 4.76 |
| Scaled MAD ( m·s$^{-1}$ ) | 8.42 | 3.84 | 3.91 |
| BIAS ( m·s$^{-1}$ ) | -1.23 | -0.98 | -0.13 |
| "y=ax" Slope | 1.17 | 0.99 | 1.01 |
| "y=ax+b" Slope | 1.12 | 0.97 | 1.00 |
| "y=ax+b" Intercept ( m·s$^{-1}$ ) | -1.16 | -1.01 | -0.12 |

## 5. Discussion

Compared and summarized by Section 4.2, the statistical results show the inter-comparison consequence of the VAL-OUC campaign of Aeolus and ground-based CDL. Because of the limited measurement height of CDL, the HLOS wind data involved in the validation, which are produced by Aeolus and CDL individually, are mainly in the PBL. It is summarized that,

for the Rayleigh-clear winds, the correlation coefficient, the standard deviation, the scaled MAD and the bias are 0.62, 7.07 m·s$^{-1}$, 5.77 m·s$^{-1}$ and -1.15 m·s$^{-1}$ respectively, while the "y=ax+b" slope and intercept are 0.96 and -1.20 m·s$^{-1}$. For the Mie-cloudy winds, the correlation coefficient, the standard deviation, the scaled MAD and the bias are 0.83, 3.15 m·s$^{-1}$, 2.64 m·s$^{-1}$ and -0.25 m·s$^{-1}$, while the "y=ax+b" slope and intercept are 0.92 and -0.33 m·s$^{-1}$.

**Table 7a. Summary of the recent comparison campaigns validation results - Rayleigh-clear**

| Campaigns/ Instruments | | R | Rayleigh-clear SD, m·s$^{-1}$ | MAD, m·s$^{-1}$ | Bias, m·s$^{-1}$ | Slope | Intercept, m·s$^{-1}$ |
|---|---|---|---|---|---|---|---|
| VAL-OUC (this study) | | 0.62 | 7.07 | 5.77 | -1.15 | 0.96 | -1.20 |
| WindVal III/ A2D (Lux et al., 2020a) | | 0.80 | 3.6 | 3.6 | 2.6 | / | / |
| WindVal III/ 2 μm DWL (Witschas et al., 2020) | | 0.95 | 4.75 | 3.97 | 2.11 | 0.99 | 2.23 |
| AVATARE (Witschas et al, 2020) | | 0.76 | 5.27 | 4.36 | -4.58 | 0.98 | -4.39 |
| AboVE-OHP (Khaykin et al., 2020) | | 0.96 | 3.2 | / | 1.5 | / | / |
| RV Polarstern cruise PS116 (Baars et al., 2020) | | / | / | 4.84 | 1.52 | 0.97 | 1.57 |
| MARA (Belova et al., 2021) | in summer | 0.82 | 5.8 | / | 0.0 | 1.1 | 0.0 |
| | in winter | 0.81 | 5.6 | / | -1.3 | 0.87 | -0.8 |
| ESRAD (Belova et al., 2021) | in summer | 0.92 | 4.5 | / | -0.4 | 1.0 | -0.5 |
| | in winter | 0.88 | 5.2 | / | -0.4 | 1.0 | -0.6 |
| WPR over Japan (Iwai et al., 2021) | Baseline 2B02 | 0.95 | 8.08 | 7.35 | 1.69 | 0.98 | 1.75 |
| | Baseline 2B10 | 0.90 | 7.89 | 7.08 | -0.82 | 0.94 | -0.74 |
| CDWL in Kobe (Iwai et al., 2021) | Baseline 2B02 | 0.98 | 6.17 | 4.92 | 0.46 | 1.05 | 0.61 |
| | Baseline 2B10 | 0.96 | 5.69 | 5.21 | -0.81 | 0.98 | -0.88 |





| CDWL in Okinawa | Baseline 2B02 | 0.93 | 6.57 | 5.68 | 1.08 | 0.99 | 1.07 |
| (Iwai et al., 2021) | Baseline 2B10 | 0.79 | 6.53 | 5.58 | -0.48 | 1.03 | -0.52 |
| GPS-RS in Okinawa | Baseline 2B02 | 0.99 | 4.55 | 4.77 | 1.00 | 0.99 | 1.00 |
| (Iwai et al., 2021) | Baseline 2B10 | 0.99 | 4.43 | 3.97 | 0.45 | 1.01 | 0.38 |
| RWP network over China (Guo et al., 2021) | | 0.94 | 4.2 | / | -0.28 | 1.01 | -0.41 |
| RS over China (Guo et al., 2021) | | 0.90 | / | / | 0.09 | 0.92 | -0.22 |

**Table 7b. Summary of the recent comparison campaigns validation results - Mie-Cloudy**

| Campaigns/ Instruments | | Mie-Cloudy | | | | | |
| --- | --- | --- | --- | --- | --- | --- | --- |
| | | R | SD, $m \cdot s^{-1}$ | MAD, $m \cdot s^{-1}$ | Bias, $m \cdot s^{-1}$ | Slope | Intercept, $m \cdot s^{-1}$ |
| VAL-OUC (this study) | | 0.83 | 3.15 | 2.64 | -0.25 | 0.92 | -0.33 |
| WindVal III/ A2D (Lux et al., 2020a) | | / | / | / | / | / | / |
| WindVal III/ 2 µm DWL (Witschas et al., 2020) | | 0.92 | 2.95 | 2.24 | 2.26 | 0.96 | 2.7 |
| AVATARE (Witschas et al, 2020) | | 0.91 | 3.02 | 2.22 | -0.17 | 1.01 | -0.21 |
| AboVE-OHP (Khaykin et al., 2020) | | / | / | / | / | / | / |
| RV Polarstern cruise PS116 (Baars et al., 2020) | | / | / | 1.58 | 0.95 | 0.95 | 1.13 |
| MARA (Belova et al., 2021) | in summer | 0.63 (Ascend); 0.72 (Descend) | 6.8 (Ascend); 6.5 (Descend) | / | 6.6 (Ascend); -0.5 (Descend) | 1.0 (Ascend); 1.3 (Descend) | 6.5 (Ascend); -2.4 (Descend) |
| | in winter | 0.73 (Ascend); 0.70 (Descend) | 5.7 (Ascend); 5.6 (Descend) | / | -1.0(Ascend); 0.9 (Descend) | 1.1 (Ascend); 1.2 (Descend) | 0.4 (Ascend); -1.2 (Descend) |
| ESRAD (Belova et al., 2021) | in summer | 0.76 (Ascend); 0.90 (Descend) | 4.7 (Ascend); 5.5 (Descend) | / | 0.5 (Ascend); 0.7 (Descend) | 0.8 (Ascend); 0.8 (Descend) | 0.5 (Ascend); 0.2 (Descend) |
| | in winter | 0.91 (Ascend); 0.85 (Descend) | 3.9 (Ascend); 5.2 (Descend) | / | 2.4 (Ascend); 0.9 (Descend) | 1.0 (Ascend); 0.9 (Descend) | 2.3 (Ascend); 0.5 (Descend) |
| WPR over Japan (Iwai et al., 2021) | Baseline 2B02 | 0.95 | 6.83 | 5.94 | 2.42 | 0.98 | 2.44 |
| | Baseline 2B10 | 0.93 | 6.47 | 5.66 | -0.51 | 0.96 | -0.44 |
| CDWL in Kobe (Iwai et al., 2021) | Baseline 2B02 | 0.98 | 4.80 | 3.55 | 1.63 | 1.05 | 1.76 |
| | Baseline 2B10 | 0.97 | 5.15 | 3.92 | 0.16 | 1.02 | 0.22 |
| CDWL in Okinawa (Iwai et al., 2021) | Baseline 2B02 | 0.97 | 3.64 | 3.76 | 2.38 | 1.01 | 2.37 |
| | Baseline 2B10 | 0.86 | 4.74 | 3.86 | -0.26 | 0.86 | -0.04 |
| GPS-RS in Okinawa (Iwai et al., 2021) | Baseline 2B02 | 0.97 | 4.52 | 4.14 | 2.15 | 0.97 | 2.07 |
| | Baseline 2B10 | 0.95 | 5.81 | 3.99 | -0.71 | 0.92 | -0.22 |
| RWP network over China (Guo et al., 2021) | | 0.81 | 6.82 | / | -0.64 | 0.99 | -0.67 |
| RS over China (Guo et al., 2021) | | 0.92 | / | / | -0.59 | 0.78 | 0.64 |

Additionally, we summarized the recent comparison campaigns from the CAL/VAL teams all over the world. The corresponding comparison results are also presented in Table 7a and Table 7b. From Table 7, the statistical parameters including correlation coefficient, SD, MAD, bias, slope and intercept of recent calibration and validation campaigns show consistent tendency and similar comparison results. The deviations among all of these studies may result from the differences



in operation strategies, spatial distances and temporal gaps and so on. In summary, considering that this study conducts the inter-comparison with the data pairs mainly in heterogeneous planetary boundary layer, the statistical results of this study are reasonable and significative.

## 6 Summary and Conclusion

To evaluate the accuracy and precision of the Aeolus-retrieved wind results, ground-based coherent Doppler wind lidars are

deployed at 17 observation stations over China for simultaneous measurements under the framework of the VAL-OUC campaign from January to December 2020. To ensure the quality of the measurement data from CDL, only wind observations with SNR>-10dB are utilized., Mie-cloudy and Rayleigh-clear wind velocities from the Aeolus L2B are selected by used with the corresponding "validity flag" of TRUE. For the comparison, the Mie-cloudy and Rayleigh-clear wind velocities with the estimated errors lower than 4 m/s and 8 m/s, respectively, are selected. Moreover, the Aeolus lowest atmospheric bins close to

the ground are removed from the comparison. In this study, the horizontal distance between the locations of CDLs and the Aeolus footprints must be less than 80 km. Since the CDL provide continuous atmospheric observations with a temporal resolution of 1 min, theoretically, there is no time difference between CDL and simultaneous Aeolus measurements. Vertical averaging of the CDL-produced wind measurements over Aeolus range bins is performed. Overall, after the strict quality control and assurance introduced above, 52 simultaneous Mie-cloudy comparison pairs and 387 Rayleigh-clear comparison

pairs from this campaign are acquired.

By the simultaneous wind measurements with CDLs and Aeolus, the Rayleigh-clear HLOS wind velocities and Mie-cloudy HLOS wind velocities from Aeolus are compared with that from CDL, respectively. All of the Aeolus-produced L2B Mie-cloudy HLOS, Rayleigh-clear HLOS and CDL-produced HLOS are compared individually. 52 measurement cases for Mie-cloudy winds could be identified for the comparison. From this statistical result, the correlation coefficient, the standard

deviation, the scaled MAD and the bias are 0.83, 3.15 $m \cdot s^{-1}$, 2.64 $m \cdot s^{-1}$ and -0.25 $m \cdot s^{-1}$ respectively, while the "y=ax" slope, the "y=ax+b" slope and the "y=ax+b" intercept are 0.93, 0.92 and -0.33 $m \cdot s^{-1}$. For Aeolus L2B Rayleigh-clear HLOS and CDL HLOS data, 387 valid observations could be used for the comparison. Accordingly, the correlation coefficient, the standard deviation, the scaled MAD and the bias are 0.62, 7.07 $m \cdot s^{-1}$, 5.77 $m \cdot s^{-1}$ and -1.15 $m \cdot s^{-1}$ respectively, while the "y=ax" slope, the "y=ax+b" slope and the "y=ax+b" intercept are 1.00, 0.96 and -1.20 $m \cdot s^{-1}$. Besides, the scatter diagrams and the

count histogram of Aeolus Rayleigh-clear HLOS according to the measurements made on ascending and descending tracks against the synchronous CDL-retrieved HLOS are plotted individually. It is found that the standard deviation and bias on ascending tracks are slightly better than that on descending tracks. Moreover, to evaluate the accuracy of Aeolus HLOS wind measurements with the baselines update, the Aeolus L2B Mie-cloudy and Rayleigh-clear HLOS wind data under Baseline 07 and 08, Baseline 09 and 10, and Baseline 11 are compared against the CDL-retrieved HLOS wind data respectively. From the

comparison results, marked misfits between the wind data from Aeolus Baselines 07/08 and wind data from CDL in planetary



boundary layers are found. After the M1 mirror temperature bias correction processor was deployed and new Rayleigh channel threshold were added, resulting in that the performances of Aeolus wind measurements under Baselines 09/10/11 are improved significantly. It has to be emphasized that the misfit may result from the contamination of Mie backscatter signal to Rayleigh backscatter signal which introduces errors to the retrieval of Rayleigh-clear HLOS velocity. Additionally, the distance between

the CDL sites and the footprint of Aeolus and the strong small-scale dynamics field may other reasons for this misfit. Finally, the statistical results of recent Aeolus wind-products calibration and validation campaigns that have been reported all over the world are summarized and compared. It is figured out that this study acquired similar results compared with other recent inter-comparison campaigns and all the comparison results show consistent tendency.

        In planetary boundary layer, the vertical velocity from convection and turbulence could influence the comparison. The

vertical velocity could an impact on the HLOS wind velocity retrieval from Aeolus. Hence, a method is described to use the vertical velocity measured with the CDL to project onto the Aeolus LOS direction and consider it for the comparison.

## Data availability

The Aeolus data are downloaded via the website of https://aeolus-ds.eo.esa.int/oads/access/collection (last accessed on 23 August 2021). The presented work includes preliminary data (not fully calibrated/validated and not yet publicly released) of

the Aeolus mission that is part of the European Space Agency (ESA) Earth Explorer Programme. This includes wind products from before the public data release in May 2020 and/or aerosol and cloud products, which have not yet been publicly released. The preliminary Aeolus wind products will be reprocessed during 2020 and 2021, which will include in particular a significant L2B product wind bias reduction and improved L2A radiometric calibration. Aerosol and cloud products will become publicly available by spring 2021. The processor development, improvement and product reprocessing preparation are performed by

the Aeolus DISC (Data, Innovation and Science Cluster), which involves DLR, DoRIT, ECMWF, KNMI, CNRS, S&T, ABB and Serco, in close cooperation with the Aeolus PDGS (Payload Data Ground Segment). The analysis has been performed in the frame of the Aeolus Scientific Calibration & Validation Team (ACVT). To get the CDL data please contact to wush@ouc.edu.cn at Ocean University of China.

## Author contributions

S. Wu contributed to the study design for intercomparison of wind measurement with Aeolus and ground based coherent Doppler lidar in the PBL over China. K. Sun, G. Dai, S. Wu and O. Reitebuch contributed to the data analyses. G. Dai and K. Sun wrote the manuscript. X. Wang (Xiaoye Wang), X. Liu, B. Liu and X. Song helped in programming. K. Sun downloaded the Aeolus data. R. Li, J. Yin and X. Wang (Xitao Wang) prepared and operated the CDLs. And all the co-authors discussed the results and reviewed the manuscript.



**Competing interests**

The authors declare that they have no conflict of interest.

**Special issue statement**

This article is part of the special issue "Aeolus data and their application". It is not associated with a conference.

**Acknowledgments**

This study has been jointly supported by the National Key Research and Development Program of China under grant 2019YFC1408001, Key Research and Development Program of Shandong Province (International Science and Technology Cooperation) under Grant 2019GHZ023 and the National Natural Science Foundation of China (NSFC) under grant 61975191 and 41905022. This work was also supported by Dragon 4 and Dragon 5 program which conducted by European Space Agency (ESA) and the National Remote Sensing Center of China (NRSCC) under grant 32296 and 59089.

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
