# Peer review of "Inter-comparison of wind measurements in the atmospheric boundary layer and the lower troposphere with Aeolus and a groundbased coherent Doppler lidar network over China"

_Atmospheric Measurement Techniques, 2021_

## Author Comment (AC2)

Response to Reviewer #2:

First of all I am very happy with this paper and the presented results. Especially the time evolution in Aeolus wind quality seen for the different baselines is impressive and valuable for other users.

Some detailed comments:

line 30:

you mention that you used data for different baselines. However, later in the paper for example on line 195 you only refer to a date for which Aeolus data was used. This makes it hard and in some cases impossible for users at a later time to know which data was used.

For example, the date of 10-May-2020 will be part of the 2nd reprocessing campaign, and therefore the same day will be available for 2 different baselines (the NRT data stream which baseline was baseline 2B10 and the reprocessing baseline 2B11).

Therefore, please state clearly which baseline and date you used if you refer to the Aeolus data for a specific date or period.

AR:
  Thank you for your suggestion.
  In the revised Fig. 7, it should be introduced firstly that the Zhangye (Gansu Province) case on 10-May-2020 is replaced by the Qingdao (Shandong Province) case for the reason that the distance between Aeolus scanning track and CDL is larger than 80 km. The description of the Baseline of this case is stated in the revised manuscript. The revised Fig. 7 and the description of the Baseline of this case are presented below:

[Figure]

"

**Figure 7. Inter-comparison of HLOS wind velocities measured with CDL and Aeolus on 16 November 2020 at Qingdao (Shandong Province), China. The left panel is the overall view of the inter-comparison result, where the red line represents the Aeolus L2B Mie-cloudy HLOS wind profile; the blue line represents the Aeolus L2B Rayleigh-clear HLOS wind profile; the black line represents the CDL-retrieved HLOS wind profile while the yellow line represents the vertical velocity corrected CDL-retrieved HLOS wind profile. The middle panel shows the partly view of the inter-comparison result, the lines are the same as the left panel. The right panel is the vertical velocity profile.**

Firstly, it should be introduced that the Aeolus L2B data of this case was produced by the processor Baseline 11."

In the revised description of Fig. 8 (shown below), the processor Baselines of data used in each case are introduced. The relevant description is present below Fig. 8 as well.
"

[Figure]

**Figure 8. Inter-comparison of HLOS wind velocities measured with CDL and Aeolus at (a) Xidazhuangke (Beijing), (b) Wuwei (Gansu Province), (c) Huludao (Liaoning Province) and (d) Qingdao (Shandong Province) on 21 January, 18 September, 15 November and 16 November 2020, respectively. The lines are the same as those of Fig. 7.**

It should be emphasized that the Xidazhuangke (Beijing) case uses the Aeolus L2B HLOS wind data on 21 January, which is from Baseline 07, while the processor of the Aeolus data in the Lanzhou (Gansu Province) case on 11 April is Baseline 08 and the processor of the Aeolus data in the Wuwei (Gansu

Province) case and Huludao (Liaoning Province) case on 15 November and 16 November are Baseline 10 and Baseline 11, respectively."

line 52:

* typo: the emotion   => the motion

AR: Thanks. The typo has been corrected in the revised manuscript.

line 68:

* typo: in the worldwide => in the world

AR: Thanks for the suggestion. This sentence has been modified as "There were some significant validation campaigns as well using airborne instruments and radiosondes (e.g., Bedka et al., 2020; Martin et al., 2020)." in the revised manuscript.

line 79:

* typo: section 5 summaries => section 5 summarizes

AR: Thanks. The typo has been corrected in the revised manuscript.

page 5: table 1: what is DBS (measurement mode)?

AR: DBS means "Doppler Beam Swinging", which is a doppler lidar measurement mode. As shown below, in the DBS measurement mode, several measurements are conducted at different $\theta$ along with one measurement in the vertical direction (Liu et al., 2019). The DBS of all CDLs deployed in the VAL-OUC has 4 tilted beams, whose $\theta$ are 0°, 90°, 180°, 270° respectively, i.e. tilted to the north, tilted to the east, tilted to the north and tilted to the west. To make it clearer, the explanation of DBS has been added below Table. 1 as "DBS*: Doppler beam swinging" in the revised manuscript.

[Figure]

**Figure 4. Schematic diagrams of DBS scan. LiDAR is placed at the origin of the Cartesian coordinate system (Liu et al., 2019).**
**Reference:**

*Liu, Z., Barlow, J., Chan, P., Fung, J., Li, Y., Ren, C., Mak, H., Ng, E: A Review of Progress and Applications of Pulsed Doppler Wind LiDARs, Remote Sens, 11 (21), 2522, https://doi.org/10.3390/rs11212522, 2019.*

line 163:

HLOS is compared to HLOS, and in section 3.2 you try to also take vertical velocity as observed by the CDL in to account. It is not yet clear to me how you apply this vertical velocity. In the example given in figure 5 is seems you apply the vertical velocity as measured at the exact time of the collocated Aeolus measurement. This I think is not correct. From figure 5 it can clearly be seen that there is a lot of variation in the vertical wind. This is probably due to convection present in the boundary layer giving sometimes updrafts and sometimes downdrafts. Aeolus accumulates over 15 to 85 km along track (in just of few seconds), so if the typical dimension of this convection is smaller than this averaging length, than the average vertical wind observed by Aeolus will be much closer to zero than what is observed by the CDL, and the net effect will be a broadening of the observed Doppler shifted spectral line only. So before applying a correction for vertical wind, first the typical size of the convective cells should be determined. Only then it can be decided if applying a correction is useful or not.

AR: The revised Fig.5 shows the moving average result of vertical velocity over 30 minutes on 16 November 2020 in Qingdao (Shandong Province), as shown below. It can be acquired from Fig. 5 that the distinct vertical velocity still exists in the moving average result. In most instances, the typical dimension of the vertical convections may be smaller than the length of the accumulation of Aeolus, but the convections whose typical dimension are similar to or larger than the Aeolus accumulation length couldn't be excluded or ignored. In this study, the ground based CDLs can't measure the vertical convection with the dimension of 15 to 85 km. Consequently, the vertical velocity correction is only conducted in the profile comparisons for the case analysis and method discussion, and not used in the statistical comparison. **The statistical errors may be introduced additionally result from the different vertical velocities of Aeolus scanning tracks and CDL sites if the method is used in the statistical comparison.** Thus, the vertical velocity correction is raised in the manuscript as a special note of Aeolus CAL/VAL campaign for further discussion.

[Figure]

**Figure 5. Vertical velocity (moving average of 30 minutes) measured on 16 November 2020 in Qingdao (Shandong Province), China. The red dashed line indicates the Aeolus transit time.**

figure 8(a) shows a case where Aeolus data from 21-Jan-2020 is used. You should mention als here that the adaptive bias correction based on ECMWF data and M1 telescope temperatures that was added with baseline 10 was not yet in place for this date (even though you do explain this in a later section). This explains the noticable bias for the Rayleigh channel winds. For this date the NRT data stream was baseline 2B07.

AR: Thanks. According to your suggestion, the Baseline description of each panel in Fig. 8 and the explanation of Fig. 8(a) has been added in the revised manuscript as "It should be emphasized that the Xidazhuangke (Beijing) case uses the Aeolus L2B HLOS wind data on 21 January, which is from Baseline 07, while the processor of the Aeolus data in the Wuwei (Gansu Province) case on 18 September is Baseline 09 and the processor of the Aeolus data in the Huludao (Liaoning Province) and Qingdao (Shandong Province) on 15 November and 16 November respectively, are both Baseline 11. It is because the adaptive bias correction based on ECMWF data and M1 telescope temperatures which was added after Baseline 09 was not yet in place for Baseline 07 that there is the noticeable bias for the Rayleigh channel winds in the Xidazhuangke (Beijing) case (Rennie et al., 2020b)."
**Reference:**
*Rennie, M., Tan, D., Andersson, E., Poli, P., Dabas, A., De Kloe, J., Marseille, G.-J. and Stoffelen, A.: Aeolus Level-2B Algorithm Theoretical Basis Document (Mathematical Description of the Aeolus L2B Processor), AED-SD-ECMWF-1025 L2B-038, V. 3.4, 124 p., https://earth.esa.int/eogateway/missions/aeolus/data (last access: 25 January 2021), 2020b.*

line 255:

you mention that you apply a data selection using "the original standard deviation" but it is not clear to me what this is or how it is defined. Please give more details.

AR: Sorry for the unclear expression. "The original standard deviation" actually is "one standard deviation" used for outlier control. In the revised manuscript, the expression of "the original standard deviation" has been replaced by "one standard deviation". The sentence is rephrased as "It should be emphasized that before these comparisons process, the outlier control is conducted firstly. The data with HLOS differences larger than one standard deviation (5.89 $m \cdot s^{-1}$ for the Mie-cloudy channel and 14.08 $m \cdot s^{-1}$ for the Rayleigh-clear channel) are removed and are not considered."

line 279: here you mention the different results for ascending and descending tracks of Aeolus data. This orbit phase depending bias has been largely solved by the adaptive bias correction based on ECMWF data and M1 telescope temperatures that was added with baseline 10. So If you only use data after 20-Apr-2020 (baseline 2B09 or newer) this should be much improved.

AR: Yes, thanks.

line 286:

you write:

"The baseline 10 dataset consists of baseline 09 data from April 2020 to October 2020 and the FM-B low bias reprocessed dataset of 2019"

this should be:

"The bias corrected dataset consists of baseline 09 data from 1 to 20 April 2020 and baseline 10 data from 20 April 2020 to 8 October 2020 and the FM-B low bias reprocessed dataset of 28 June 2019 to 31 December 2019."

AR: Thanks. This sentence has been replaced in the revised manuscript.

summary tables 7a and 7b do not really belong to this study. It raises much more questions, i.e. are alle these studies just looking at the boundary layer as well? (I guess not). How much data did they use? What extra quality controls did they implement, etc. My feeling is that comparing CalVal results from many teams should be published in another report or paper and not here.

AR: Yes, Table. 7a and Table 7b indeed not belong to this study. The purpose of the comparison between the statistical results of this study and those of other studies is that, though this study conducts the inter-comparison with the data pairs mainly in heterogeneous atmospheric boundary layer, compared with the published inter-comparison results, the statistical results of this study are reasonable and significant. Therefore, we insist that the existence of Table. 7a and Table 7b is necessary and reasonable.

---

## Author Response (AR1)

Dear Editor and Reviewers,

Thank you for handling and reviewing our manuscript. We greatly appreciate the substantial amount of time and effort that you dedicated to this review process.

We have revised the manuscript according to your comments point-to-point and the responses are presented below.

It should be introduced that the serial numbers of the figures and the tables are changed in the revised manuscripts.

Many thanks and best regards.

Songhua Wu On behalf of the co-authors Response to Reviewer #1:

**General comments:**

Based on one year of simultaneous wind measurements acquired from 17 Doppler wind lidars across China, this manuscript by Wu et al. conducted a comprehensive comparison study against Aeolus wind products. Overall, this topic fits well the AMT. The instruments and data are reliable, and the analysis methods are scientifically sound. In my opinion, the manuscript is well organized except for some typos and grammar errors. The comparison results are of great importance to better understand the performance of Aeolus wind products in China, even though the measurements are obtained from coherent Doppler lidar (with 1550 nm wavelength) over China. However, before the manuscript can be recommended for acceptance for publication, I have several suggestions and comments here that need to be addressed.

Specific comments:

 Section 2.2.2: Line 128 says "The measurement heights selected for comparison are 50 m, 100m." Nevertheless, I only see the comparison results at 50 m in Figure 2. I am curious why not showing the results at 100 m AGL? 2 only compares the wind speed without considering the wind direction. I suggest the authors compare the u-component wind.

**AR:**

Sorry about the misleading. We actually compared both the heights at 50 m and 100 m for the wind speed and direction (as shown below). From the comparison, the slopes, offsets, correlation coefficients, standard deviations and BIAS all have same consistence. We think the comparison results at 50 m is enough to demonstrate the stability and the precision of Wind3D 6000 and WindMast PBL, thus we didn't plan to present the comparison results at 100 m in the manuscript, which might not make more sense. Besides, the manuscript will be tedious if we do that. We modified the sentences as "The measurement heights selected for comparison are 50 m, 100m. Figure 3 shows the comparison results at 50 m, which are wind speed and wind direction for Wind3D 6000 and WindMast PBL, respectively." in the revised manuscript.

Thank you for your suggestion. The wind direction comparison results at 50 m are shown in Fig. 3 in the revised manuscripts, which are presented as below as well:

Figure 3. Evaluation tests of (a), (b) Wind3D 6000 and (c), (d) WindMast PBL performance by comparing their measurements against the conventional wind measurements with mast mounted cup anemometers and wind vanes.

Besides, the comparison results between Wind3D 6000/WindMast PBL and cup anemometers/wind vanes at 100 m AGL are shown below for the further knowledge.

---

## Author Response (AR2)

Response to Reviewer #1:

Reviewer comments:

The authors have appropriately addressed my concerns, for which i appreciate it very much. I have only one minor comment for the authors' consideration. I noticed that the altitude range analyzed here (the number of data pairs in Figure 9) are mostly higher than 1.5 km, which is generally thought as the average atmospheric boundary height. Hence, "atmospheric boundary layer" in the title can be revised to "lower troposphere".

AR: Thank you for your suggestion. In the title, "the atmospheric boundary layer" has been revised to "the atmospheric boundary layer and the lower troposphere". Besides, we checked the entire manuscript and made the same modifications throughout the manuscript.